# Resistance to Helium Bubble Formation in Amorphous SiOC/Crystalline Fe Nanocomposite

**DOI:** 10.3390/ma12010093

**Published:** 2018-12-28

**Authors:** Qing Su, Tianyao Wang, Jonathan Gigax, Lin Shao, Michael Nastasi

**Affiliations:** 1Department of Mechanical and Materials Engineering, University of Nebraska-Lincoln, Lincoln, NE 68583-0857, USA; mnastasi2@unl.edu; 2Department of Nuclear Engineering, Texas A&M University, College Station, TX 77843-3128, USA; wty19920822@tamu.edu (T.W.); gigaxj@tamu.edu (J.G.); lshao@tamu.edu (L.S.); 3Nebraska Center for Energy Sciences Research, University of Nebraska-Lincoln, Lincoln, NE 68583-0857, USA; 4Nebraska Center for Materials and Nanoscience, University of Nebraska-Lincoln, Lincoln, NE 68588-0298, USA

**Keywords:** radiation tolerant materials, amorphous silicon oxycarbide, nanocrystalline Fe, composite, interface

## Abstract

The management of radiation defects and insoluble He atoms represent key challenges for structural materials in existing fission reactors and advanced reactor systems. To examine how crystalline/amorphous interface, together with the amorphous constituents affects radiation tolerance and He management, we studied helium bubble formation in helium ion implanted amorphous silicon oxycarbide (SiOC) and crystalline Fe composites by transmission electron microscopy (TEM). The SiOC/Fe composites were grown via magnetron sputtering with controlled length scale on a surface oxidized Si (100) substrate. These composites were subjected to 50 keV He+ implantation with ion doses chosen to produce a 5 at% peak He concentration. TEM characterization shows no sign of helium bubbles in SiOC layers nor an indication of secondary phase formation after irradiation. Compared to pure Fe films, helium bubble density in Fe layers of SiOC/Fe composite is less and it decreases as the amorphous/crystalline SiOC/Fe interface density increases. Our findings suggest that the crystalline/amorphous interface can help to mitigate helium defect generated during implantation, and therefore enhance the resistance to helium bubble formation.

## 1. Introduction

The combination of irradiation defects and helium (He) lead to a microstructural evolution of bubbles, cavities and voids, which ultimately lead to the degradation of mechanical properties in first-wall materials as well as fuel cladding in fission nuclear reactors [1,2]. For example, formation of He bubbles at grain boundaries of austenitic stainless steel has been found to occur even at very low overall He concentrations, causing deleterious effects such as swelling and embrittlement [3,4].

Over past decades, extensive researches have been conducted to understand the behavior of inert gases such as helium in pure elemental metals [5,6]. The development of radiation tolerant composite materials via the introduction of interfaces, phase boundaries, and grain boundaries has also been discussed in a number of investigations [7,8,9,10]. For instance, oxide dispersion strengthened (ODS) steels, which contain a high volume fraction of metal/nanoscale oxides interfaces, has shown that nanoscale precipitates can promote the recombination of radiation-induced point defects, and therefore mitigate He bubbles formation [1,11,12]. Similar to the interface effect due to precipitates in ODS steels, the introduction of well-controlled nanoscale metallic interfaces (e.g., interfaces between face-centered cubic and body-centered cubic materials) have also been shown to be efficient for trapping He and mitigating the onset of He bubble formation [9,13,14].

While the above discussion on interface design strategies have shown that it is possible to delay the deleterious effects of He, recent studies have shown that in some materials it is possible to avert helium bubble formation entirely by continually removing it as it is implanted [15]. Amorphous SiOC, a new class of superior radiation tolerant materials, has shown very good steady-state irradiation properties [16]. Previous studies have demonstrated that amorphous SiOC alloys are stable under irradiation, sustaining their glassy states over a wide range of irradiation conditions [17,18,19]. More interestingly, implanted He atoms were found to diffuse out of the SiOC matrix as fast as it was implanted, even at liquid nitrogen temperatures, resulting in time-invariant structure and properties. In addition, amorphous SiOC can be paired with a crystalline metal component such as Fe to form a composite with enhanced thermal, mechanical and irradiation properties [20,21]. However, at present the properties of SiOC/Fe composites under helium implantation, remain virtually uncharted. Similar to what has been observed in metal/metal nano-composites and ODS steels, we hypothesize that the crystalline/amorphous interfaces in the SiOC/Fe composite films are able to facilitate vacancy and interstitial recombination, therefore, resulting in enhanced helium bubble formation resistance [10,22,23]. In this work, we investigated and compared the He implantation responses of pure Fe films and SiOC/Fe composites with controlled length scale for the first time. The results serve to better understand the role of SiOC/Fe amorphous/crystalline interfaces on helium management and defect mitigation in harsh environments.

## 2. Materials and Methods 

Magnetron sputtering was used to synthesize SiOC/Fe multilayer films with controlled individual layer thicknesses. Direct current (DC) magnetron sputtering was used to deposit α-Fe layers, while radio frequency (RF) sputtering was used to synthesize amorphous SiOC layers from co-sputtering SiO_2_ and SiC targets. Prior to depositions, a base pressure of 9.2 × 10^−6^ Pa or lower was obtained and the typical argon partial pressure during sputtering was ~0.65 Pa. The thickness of the pure α-Fe film and two kinds of SiOC/Fe multilayered films was ~1 μm. In thick SiOC/Fe multilayer films, the thickness of Fe and SiOC layers were 80 and 60 nm, respectively, while the thickness of Fe and SiOC layers for thin SiOC/Fe multilayer films were 16 and 12 nm. The roughness of typical SiOC and Fe layer ranges from 3 to 5 nm, as suggested by previous X-ray reflectivity experiment [24]. All targets including SiO_2_ (purity 99.995%), SiC (purity 99.5%) and Fe (purity 99.95%), were obtained from AJA International, Inc. (North Scituate, MA, USA)

The pure SiOC film, pure Fe film and SiOC/Fe multilayers were subject to 50 keV He ions implantation at room temperature. Stopping and Range of Ions in Matter (SRIM)-2008 software was used to simulate depth profiles of implanted ion concentration and irradiation damage using the ion distribution and quick calculation of damage option [25]. The SiOC/Fe nanolaminate was treated as a uniformly distributed amorphous target material for the purpose of the simulations. The nominal composition for thick and thin SiOC/Fe nanolaminates are Fe_13.3_Si_3_O_4_C_3_. The assumed displacement energies for Si, O, C and Fe are 15, 28, 28 and 40 eV, respectively. Rutherford backscattering spectrometry and X-ray reflectivity results suggest the SiOC films possess chemical composition of Si-30%, O-40%, C-30% and density of 2.2 g/cm^3^ [24]. The density of Fe layers is 6.92 g/cm^3^, approximately 14% lower than that of pure Fe target due to shadowing effects during sputtering process [26]. The base pressure during He implantation was better than 5 × 10^−4^ Pa. To obtain a 5 at% He peak concentration, fluences of 6.8 × 10^16^, 6.5 × 10^16^ and 7.0 × 10^16^ ion/cm^2^ was implanted into pure SiOC film, pure Fe film and SiOC/Fe multilayers, respectively. The 5 at% He peak was chosen in order to visualize the He bubbles in Fe layers. Because the He concentration profile is near-Gaussian and therefore, the implant concentration varied between 0 and 5 at% as a function of depth. It allowed to investigate He bubble formation in this range of implant concentration. The beam spot size was 8 mm × 10 mm. The fluence variation within the beam spot was typically within ±10%. The fluence was measured by monitoring the charge collection on the target. The target was biased during the irradiation to suppress the error caused by secondary electrons. Such a setup has shown good accuracy in fluence determination, of uncertainty of <15%, based on previous testing from secondary ion mass spectrometry analysis of various implants in Si and Fe substrates. The cross-sectional microstructure of SiOC/Fe multilayers and pure Fe films before and after implantation was characterized by TEM. The cross-sectional TEM specimen was prepared by conventional dimple and grinding followed by ion-milling. Low energy (3.5 keV) and low angle (5°) were selected to reduce the ion milling damage. A FEI Tecnai G2 F20 TEM (FEI, Hillsboro, OR, USA) was used to investigate the microstructure of these films before and after He implantation. The typical TEM operation voltage was 200 kV. The thickness of TEM foils of all specimens were determined by Electron energy loss spectroscopy (EELS) log-ratio technique, as described in the literature [27]. The thicknesses of analyzed pure Fe film, thin and thick SiOC/Fe multilayers were 68.2 ± 6.1, 74.1 ± 8.5 and 52.8 ± 4.2 nm. The detailed microstructure analysis of as-deposited SiOC film, Fe film and SiOC/Fe multilayers films can be found in previous references [28,29].

## 3. Results

### 3.1. SRIM Simulation 

To obtain He doping within these films, 50 keV He ions were selected for ion implantation. The depth profiles of radiation damage in units of displacement per atom (dpa) and helium concentration in the SiOC, Fe and Fe/SiOC films are shown in Figure 1a–c, respectively. The simulations, as shown in Figure 1, implies that all films were subjected an implantation that would result in a 5 at% peak helium concentration, assuming all the implanted He was retained. In addition to helium implantation, the He^+^ irradiation results in maximum 2.6 dpa in the pure SiOC films and ~3 dpa in both the Fe and the SiOC/Fe composite films.

### 3.2. He Implantation in Fe and SiOC Films

The cross-sectional TEM images of the pure Fe film after 5 at% He implantation are shown in Figure 2a,b. Similar to the as-deposited film, the implanted Fe film exhibits a columnar structure. The corresponding selective area diffraction (SAD) pattern shown in the inset of Figure 2a suggests the implanted Fe film still retained a bcc structure. An under-focused cross-sectional TEM micrograph of the implanted Fe film is shown in Figure 2b. The micrograph was collected in the He peak region, which is 233 nm underneath the surface. A high density of He bubbles is observed within the columnar grains as well as along grain boundaries. To estimate the average He bubble size, cross-sectional TEM micrographs were taken at an under-focus distance of 400 nm. The average bubble diameter for the Fe film is 1.2 ± 0.1 nm, in comparison to a 1.1 ± 0.1 nm bubble size in bulk Fe [8]. In contrast to the high density of He bubbles in implanted Fe films, no helium bubbles (>1 nm) were observed in pure SiOC film after 5 at% He implantation, Figure 2c,d. This result was consistent with previous finding that He atoms in SiOC remain in solution and are able to outgas from the material via atomic-scale diffusion [15,28]. In addition, the irradiation does not lead to any void formation, element segregation or crystallization throughout the SiOC film. 

### 3.3. He Implantation in SiOC/Fe Multilayers

To investigate the amorphous/crystalline SiOC/Fe interface effect on helium bubble formation in SiOC/Fe composite films, He^+^ implantations were also conducted for these composite films at room temperature. Figure 3a,b shows low magnification TEM micrographs from thick and thin Fe/SiOC films after 5 at% peak He implantation. Fresnel contrast is used to examine the helium formation which are presented as dark dots surrounded by a bright fringe for the over-focus condition and bright dots surrounded by a dark fringe in the under-focus condition. The cross-sectional micrographs of under-focused and over-focused thin Fe/SiOC films at the He peak concentration regions are shown at Figure 3c,d, respectively. In order to compare the helium behavior between thin and thick Fe/SiOC films, Figure 3e,f presents typical under-focus and over-focus images for thick multilayer specimen. One of most important observations is the presence of helium bubbles in Fe layers but not in SiOC layers. For example, these helium bubbles are revealed via Fresnel contrast in both the under-focus condition (Figure 3c,e) and the over-focus condition (Figure 3d,f) in Fe layers. It is also interesting to note no helium bubbles were observed along the SiOC/Fe interface. Scanning transmission electron microscopy (STEM) was utilized under a high-angle annular dark field (HAADF) condition to further examine the composition profile of thin and thick SiOC/Fe multilayers after 5 at% He implantation. The corresponding STEM images of Figure 3a,b are shown as Figure 3g,h, respectively. These data indicate that the layer structure is maintained and SiOC/Fe interface remain quite sharp after He implantation. 

### 3.4. Depth Profile of He Bubble Density

The density of He bubbles at different depths in SiOC/Fe multilayers and Fe films was measured via extensive cross-sectional TEM analysis. Figure 4a,b summarizes that the bubble density profile in SiOC/Fe multilayers and in pure Fe film, respectively. It is interesting to note that the bubble density depth profiles in all films coincided with simulated He concentration profiles (solid line) superimposed in the same figure. In SiOC/Fe multilayers, the helium bubble density in Fe layers increases continuously and approaches a maximum at an implantation depth of ~310 nm, while the helium bubble density in pure Fe film reaches maximum at a depth of ~240 nm. The average bubble diameters for the Fe layers in thick and thin composite were 1.2 ± 0.1 and 1.2 ± 0.2 nm, respectively. No or very few change in bubble diameters was observed with depth. It can be seen that, under the same effective He implanted concentration, the helium bubble density in pure Fe films is approximately twice as that in thick SiOC/Fe multilayer sample. In addition, a lower helium bubble density profile is observed in thin SiOC/Fe multilayer sample. We also attempted to estimate the threshold concentration needed for the formation of detectable He bubbles in pure Fe films and SiOC/Fe multilayers. As shown in Figure 4a, a minimum He concentration to visualize bubbles is obtained from the intersections of the vertical lines with the SRIM simulation. Compared to the low threshold concentration in pure Fe film (<0.1 at%), the threshold concentration is approximately 0.82 at% and 1.95 at% in thick and thin SiOC/Fe multilayer sample, respectively, which represents an order of magnitude improvement for the composites. 

## 4. Discussion

Previous work has shown that radiation induced defects prefer to migrate to the interfacial regions that act as effective defect sinks [30,31]. The SiOC/Fe amorphous/crystalline interface density in SiOC/Fe 12/16 nm nanolayers is 10 times higher than that in SiOC/Fe 60/80 nm nanolayers. Therefore, it appears that a high density of amorphous/crystalline interfaces also facilitate defect removal. Recent molecular dynamic (MD) simulation studies of helium defect properties in bcc iron showed that helium interstitials as well as He interstitial clusters are quite mobile [32,33]. The mobility of helium interstitials or He interstitial clusters will decrease if they are trapped by vacancies. These vacancies could be intrinsic vacancies, irradiation induced vacancies, or vacancies created via kick-out of a self-interstitial atom by He clusters. The smaller helium bubble density observed in the SiOC/Fe multilayer system suggests that there is a lower concentration of vacancies in the Fe layers of SiOC/Fe composite specimen compared to that in the pure Fe layer during implantation. In both implanted Fe films and SiOC/Fe multilayers, He bubbles have an average diameter of ~1 nm and the bubble size distributions is quite narrow. As suggested by cross-section TEM images, the average bubble size depends very little on Fe layer thickness. The reduction in He bubble density in thin SiOC/Fe multilayers compared to that in thick SiOC/Fe multilayers and Fe films indicates that the vacancy concentration must have been dramatically reduced. All these results imply that the amorphous/crystalline interfaces act as efficient defect sinks to promote interstitials and vacancies recombination and to lessen helium bubble formation in the Fe layers.

Another interesting observation is that no helium bubbles were observed at the SiOC/Fe interfaces. Considering the fast helium diffusivity in SiOC, the helium atoms most likely eventually diffuse out of the system once they reach the SiOC/Fe interfaces. Therefore, the amount of helium that could diffuse out the Fe layers depends on whether helium atoms or helium clusters are able to reach the interface region. This suggests that by refining the layer thickness, it may be possible to obtain a composite free of helium bubbles. 

## 5. Conclusions

In conclusion, we examined the effect of helium implantation on the Fe/SiOC composite system after 5 at% implantation at room temperature. By introducing amorphous/crystalline SiOC/Fe interfaces, the onset concentration needed to visualize helium bubbles in the Fe layers of SiOC/Fe composites is ten times lower than needed in pure Fe films. In addition, the helium bubble density decreases as interface density increases. All these observations suggest that the Fe/SiOC crystalline/amorphous interfaces act as efficient defects sinks, promoting interstitials and vacancies recombination and mitigating helium bubble accumulation. 

## Figures and Tables

**Figure 1 materials-12-00093-f001:**
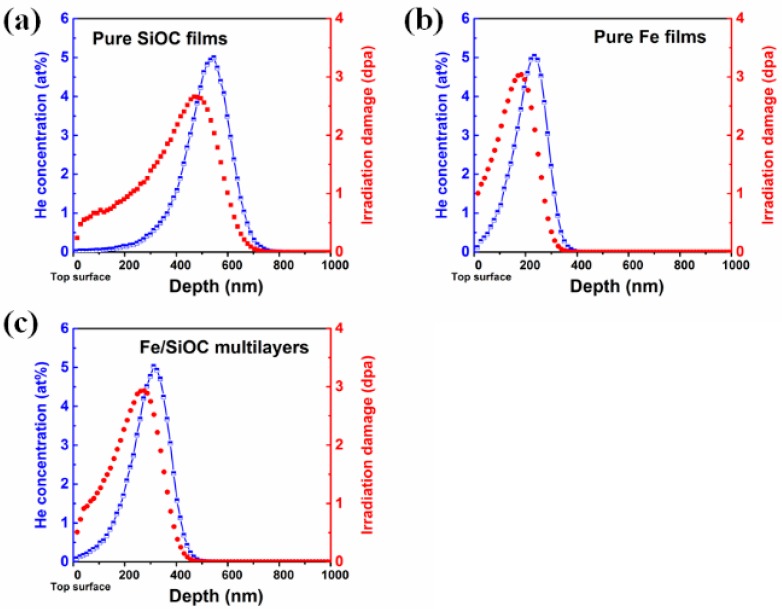
The simulated depth profile of radiation damage and helium concentration in (**a**) pure SiOC film, (**b**) pure Fe film and (**c**) Fe/SiOC multilayers. The peak concentration for all films are 5 at%.

**Figure 2 materials-12-00093-f002:**
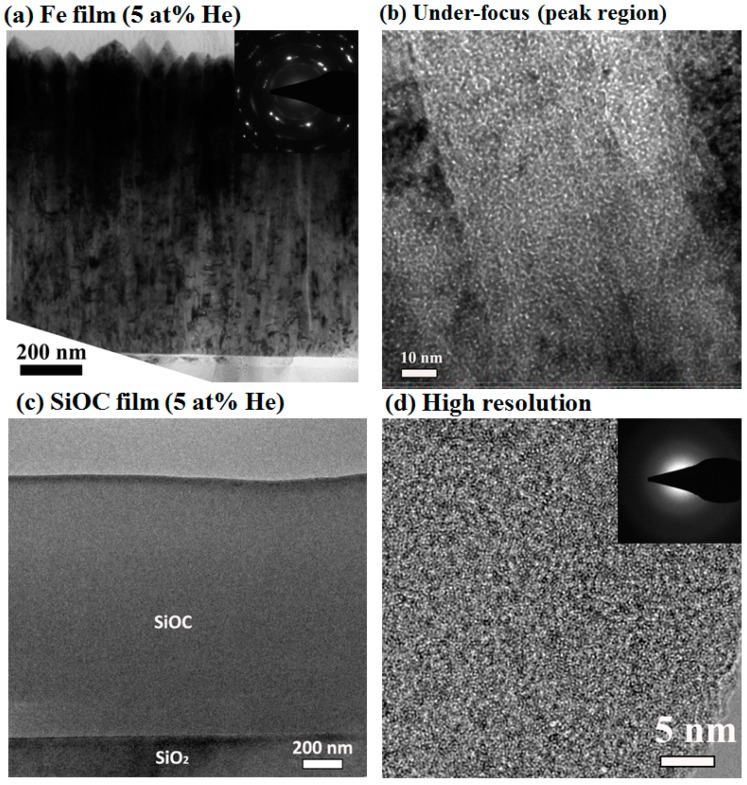
The typical cross-sectional TEM images of (**a**) pure Fe film and (**c**) pure SiOC film after 5 at% He implantation. The high-resolution TEM images of He peak regions are shown as (**b**,**d**), respectively.

**Figure 3 materials-12-00093-f003:**
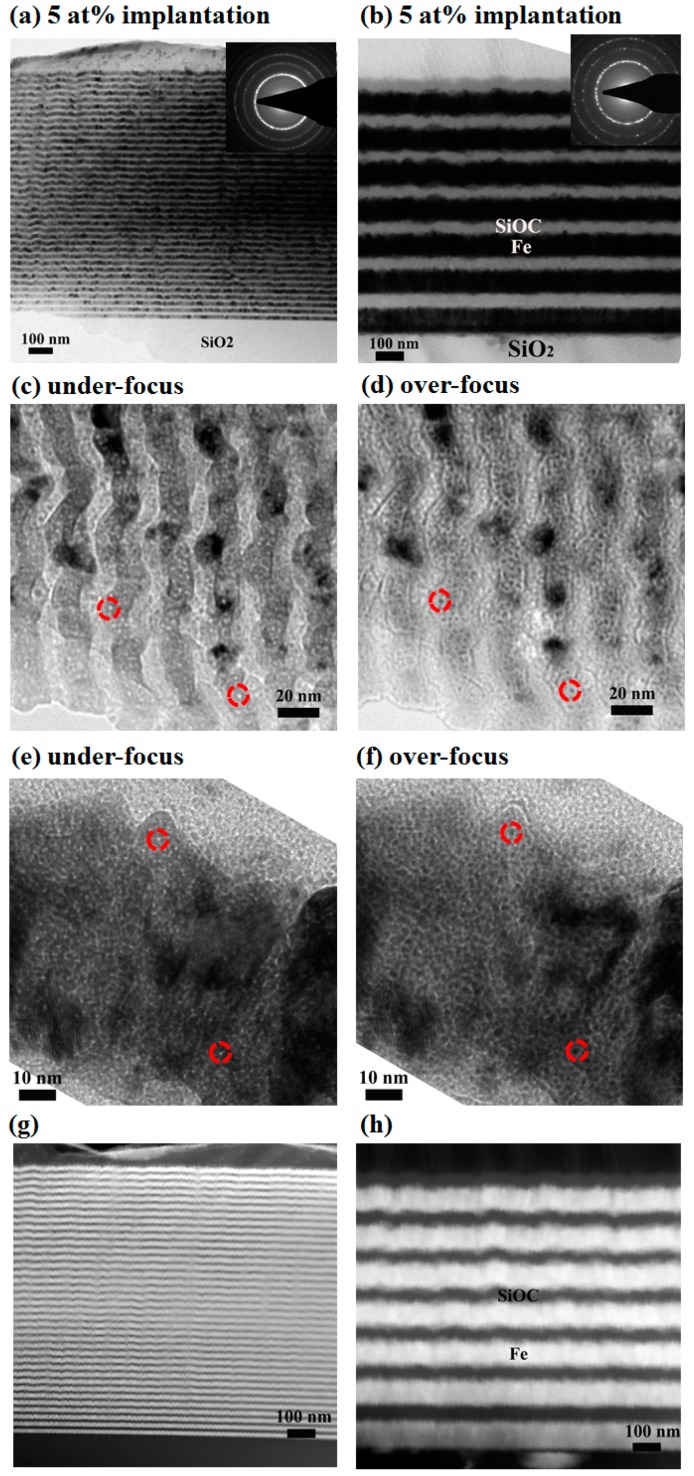
The typical cross-sectional TEM images of (**a**) thin, (**b**) thick Fe/SiOC multilayers after 5 at% He implantation. The He peak damage regions of thin and thick Fe/SiOC multilayers are taken at under-focus (−400 nm) and over-focus (+400 nm) condition, respectively, shown as (**c**–**f**). The corresponding scanning transmission electron microscopy (STEM) images are present as (**g**,**h**), respectively.

**Figure 4 materials-12-00093-f004:**
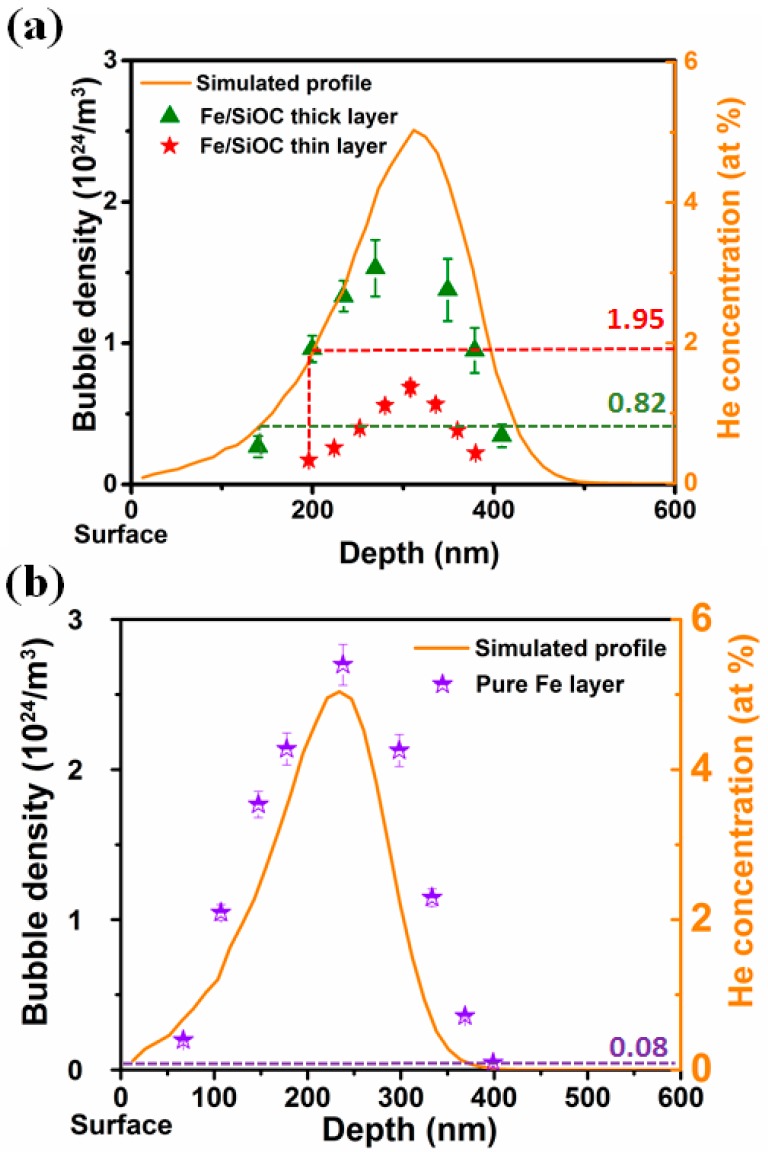
He bubble density (scattered points) as a function radiation depth in (**a**) Fe/SiOC multilayer and (**b**) pure Fe films. Solid line in each figure is simulated helium depth profile. The dashed line was drawn to help estimate the threshold concentration for the formation of detectable He bubbles. Bubble density in Fe layers of Fe/SiOC multilayers is significant lower than that in pure Fe film.

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
