# Peer review of "Resistance to Helium Bubble Formation in Amorphous SiOC/Crystalline Fe Nanocomposite"

_materials, 2018, doi:10.3390/ma12010093_

Round 1
Reviewer 1 Report
General comment and recommendation:
The manuscript describes the formation of a helium bubbles in SiOC and Fe composites and compares these results to the bubble formation in pure Fe and SiOC. The helium bubble formation in SiOC and Fe composites is reduced compared to pure Fe. It is concluded that the Fe/SIOC interface enhances the resistance of the composite material to helium bubble formation.
The manuscript describes a very interesting and important effect that might help to develop radiation hardened materials. The experiments and measurements were performed thoroughly, and the conclusions are supported by the presented data. The manuscript requires only minor modifications.
Specific comments:
The English language is understandable, but could be improved. A revision by a native English speaker is recommended.
Page 2, line 69-70: “The thickness of the pure α‐Fe film…” If anything is known about the oxygen and/or impurity concentration in the deposited film, then this information could be added.
Page 2, lines 70 – 72: How many multilayers were deposited?
Page 2, lines 70 – 72 and Figs. 3a) and 3g): The multilayers show some roughness. If any quantitative information about the roughness is available, then it could be added.
Page 2, lines 77 – 78: “The Fe/SiOC nanolaminate was treated as a uniformly distributed amorphous target material…” The composition of this uniformly distributed material should be given for clarity.
Page 2, line 79: “The displacement energies for Si, O, C and Fe are 15, 28, 28 and 40 eV, respectively.” For clarity this statement should be changed to “The assumed displacement energies for Si, O, C and Fe are 15, 28, 28 and 40 eV, respectively.”
Page 2, line 81: “…and density of 2.2 g/cm3” How does this density compare to the theoretical density of SiOC?
Page 2, Section 2: Is anything known about the density of the Fe film?
Page 2, lines 82 - 83: “… fluences of 6.8×1016, 83 6.5×1016 and 7.0×1016 ion/cm2 was implanted…” What was the size of the implantation spot? What was the homogeneity of the implantation? How was the fluence measured?
Figure 1: For clarity it would be better to draw the x-axes only until 800 or 900 nm.
Page 3, line 116: “…no helium bubbles were observed…” It should be discussed shortly, if very small bubbles not visible in TEM could be present. The smallest visible bubble diameter should be given.
Page 6, lines 145 – 160: Some information about the diameters of the bubbles shown in Fig. 4(a) should be given. Do the bubble diameters change with depth?
Figure 4: The dashed lines are explained in the text, but should be explained also in the figure caption.
Author Response
Please see details from the attachment.

Reviewer 2 Report
The manuscript is devoted to an interesting problem, and obtained results are interesting as well. However, several issues must be fixed before publishing.
1. The thickness of Fe and SiOC layers for thin and thick SiOC/Fe films are described in section 2, lines 70 and 71. Why these thicknesses were chosen?
2. Simulation with SRIM software was performed using “ion distribution and quick calculation of damage option” (Kinchin-Pease model). The opinion of the author of SRIM software on this option can be found in “TRIM Help”: “This option should be used if you don't care about details of target damage or sputtering”.
Applicability of this model to the calculation of irradiation damage (Figure 1) is not proven in the manuscript. Neglecting collision cascades for light ions in this energy range seems to be not crucial, but not sufficiently argumented in the text.
3. Line 77-78: “The Fe/SiOC nanolaminate was treated as a uniformly distributed amorphous target material for the purpose of the simulations”. Layered target can be easily created in SRIM software, so, it is not clear, why uniform FeSiOC mixture was used instead of multilayers.
4. Line 82: “To obtain a 5 at% He peak concentration…” Why 5 at% He peak concentration? Is it possible to see He bubbles if the He peak concentration is 3 at% or 7 at%? Please specify the reason for chosen concentration.
5. Line 96: “…units of dpa…” Abbreviation “dpa” is not explained.
6. Lines 125-127, “Room temperature” is mentioned twice for SiOC/Fe composite films. The temperature is an important parameter, but it was already described in “Materials and Methods”, and not repeated for Fe and SiOC films. Why repeated in this case?
7. The description of figure 3 is too short, when the figure itself contains a lot of images. The arrangement of images in figure is not optimal for understanding (for my eye). Thin at left and thick at right, then thin at left and right, then both thick and then thin at left and thick at right again. Word “respectively” is used in figure caption, but when describing 2x2 array it is not very useful.
8. In figures 3(e) and 3(f) I can’t find contrast from bubbles in Fe layer. It is not surprising at such magnification if the size of the bubble is of about 1 nm. In figures (c) and (d) mean diameter of a contrast from bubbles is of about 3 pixels, so in figures 3(e) and 3(f) it is expected to be less than 1 pixel. I would recommend using other images at high magnification. It would be very useful to mark one or two bubbles in image using arrows.
9. Reference [16] (J Nucl Mater 2015, 461, (0), 200-205) is repeated: [24] (J Nucl Mater 2015, 461, 200-205)
Author Response
Please disclose the response from the attachment. Thanks.

Round 2
Reviewer 2 Report
The manuscript is improved after revision, however some issues are still present.
1) Line 86-87: "To obtain a 5 at% He peak concentration, fluences of ...... ion/cm2 was implanted". For me this sentence means that the author chose ion fluences to obtain 5 at%, because this concentration is required for some purpose, which is not explained in the text. There is an explanation of the idea in authors reply to this question from my previous report, but it is not included in reviced manuscript.
2) The description of Fresnel contrast from bubbles "as dark dots surrounded by a bright fringe for the over‐focus condition and bright dots surrounded by a dark fringe in the under‐focus condition" is repeated twice in one paragraph: lines 140-141 and lines 147-148.
Author Response
Please disclose the response from the attachment.
